# *RET* C611Y Germline Variant in Multiple Endocrine Neoplasia Type 2A in Denmark 1930–2021: A Nationwide Study

**DOI:** 10.3390/cancers17030374

**Published:** 2025-01-23

**Authors:** Anders Würgler Hansen, Peter Vestergaard, Morten Møller Poulsen, Åse Krogh Rasmussen, Ulla Feldt-Rasmussen, Mette Madsen, Rune Weis Næraa, Dorte Hansen, Katharina Main, Henrik Baymler Pedersen, Stefano Christian Londero, Lars Rolighed, Christoffer Holst Hahn, Klara Bay Rask, Christian Maare, Heidi Hvid Nielsen, Mette Gaustadnes, Maria Rossing, Pernille Hermann, Jes Sloth Mathiesen

**Affiliations:** 1Department of ORL Head & Neck Surgery and Audiology, Odense University Hospital, 5000 Odense, Denmark; jes_mathiesen@yahoo.dk; 2Steno Diabetes Center North Denmark, Aalborg University Hospital, 9000 Aalborg, Denmark; p-vest@post4.tele.dk (P.V.); mette.madsen@rn.dk (M.M.); 3Department of Clinical Medicine, Aalborg University Hospital, 9000 Aalborg, Denmark; 4Department of Endocrinology, Aalborg University Hospital, 9000 Aalborg, Denmark; 5Department of Endocrinology and Internal Medicine, Aarhus University Hospital, 8200 Aarhus, Denmark; morpouls@rm.dk; 6Department of Nephrology and Endocrinology, Copenhagen University Hospital, 2100 Copenhagen, Denmark; aase.krogh.rasmussen@regionh.dk (Å.K.R.); ulla.feldt-rasmussen@regionh.dk (U.F.-R.); 7Department of Pediatrics and Adolescent Medicine, Aalborg University Hospital, 9000 Aalborg, Denmark; 8Department of Pediatrics, Aarhus University Hospital, 8200 Aarhus, Denmark; runenaer@rm.dk; 9Department of Pediatrics, Odense University Hospital, 5000 Odense, Denmark; dorte.hansen4@rsyd.dk; 10Department of Growth and Reproduction, Copenhagen University Hospital, 2100 Copenhagen, Denmark; katharina.main@regionh.dk; 11Department of ORL Head & Neck Surgery, Aalborg University Hospital, 9000 Aalborg, Denmark; hebp@rn.dk; 12Department of ORL Head & Neck Surgery, Aarhus University Hospital, 8200 Aarhus, Denmark; stelon@rm.dk (S.C.L.); larsroli@rm.dk (L.R.); 13Department of ORL Head & Neck Surgery, Copenhagen University Hospital, 2100 Copenhagen, Denmark; christoffer.holst.hahn.01@regionh.dk (C.H.H.); klara.bay.rask@regionh.dk (K.B.R.); 14Department of Oncology, Copenhagen University Hospital-Herlev and Gentofte, 2730 Herlev, Denmark; christian.maare@regionh.dk; 15Department of Clinical Biochemistry, Zealand University Hospital, 4000 Roskilde, Denmark; hhne@regionsjaelland.dk; 16Department of Molecular Medicine, Aarhus University Hospital, 8200 Aarhus, Denmark; mette.gaustadnes@clin.au.dk; 17Department of Genomic Medicine, Copenhagen University Hospital, 2100 Copenhagen, Denmark; caroline.maria.rossing@regionh.dk; 18Department of Clinical Medicine, Faculty of Health and Medical Sciences, University of Copenhagen, 2100 Copenhagen, Denmark; 19Department of Endocrinology, Odense University Hospital, 5000 Odense, Denmark; pernille.hermann@rsyd.dk; 20Department of Clinical Research, University of Southern Denmark, 5000 Odense, Denmark

**Keywords:** multiple endocrine neoplasia type 2, RET, C611Y, genotype, phenotype

## Abstract

Multiple endocrine neoplasia type 2A (MEN 2A) is a rare hereditary cancer syndrome characterized by medullary thyroid carcinoma (MTC), pheochromocytoma (PHEO), primary hyperparathyroidism (PHPT), cutaneous lichen amyloidosis (CLA), and Hirschsprung’s disease. The syndrome is caused by pathogenic variants in the rearranged during transfection (*RET*) proto-oncogene. Phenotypic data on the pathogenic *RET* variant C611Y remain sparse. We therefore conducted a Danish nationwide study of all cases (n = 128). The C611Y variant is associated with a very high penetrance of MTC and a low penetrance of PHEO and PHPT. CLA and Hirschsprung’s disease were not observed. MTC seems moderately aggressive in comparison to other *RET* variants. Overall survival may be comparable to that of the general population.

## 1. Introduction

Multiple endocrine neoplasia type 2A (MEN 2A) (OMIM #171400) is a rare hereditary cancer syndrome with a point prevalence of 13–24 per million [1,2]. The syndrome predisposes carriers to the development of medullary thyroid cancer (MTC), pheochromocytoma (PHEO), primary hyperparathyroidism (PHPT), cutaneous lichen amyloidosis (CLA), and Hirschsprung’s disease [3,4,5].

MEN 2A is inherited in an autosomal dominant pattern. The syndrome is caused by activating germline mutations in the rearranged during transfection (*RET*) proto-oncogene. Shortly after this was discovered in 1993 [6,7], strong genotype–phenotype correlations were observed by the International *RET* Mutation Consortium [8,9]. Since then, solid clinical risk profiles have been established for various *RET* variants [10,11,12,13,14,15,16,17,18]. However, precise phenotypic data on carriers of variants in codon 611 and specifically for carriers of the C611Y variant remain sparse. Thus, the International *RET* Exon 10 Consortium investigated 50 carriers of codon 611 variants, 15 of whom harbored the C611Y variant [10]. Meanwhile, a Chinese study reported on a pedigree comprising 17 C611Y carriers [19]. The largest study of C611Y carriers to date, a German single center study, included 42 carriers of codon 611 variants, with 19 harboring the C611Y variant [20]. Likely due to a founder effect, the C611Y variant is the most common in Denmark, accounting for 128 of 216 MEN 2A cases [21,22,23]. This provides the perfect setting for a nationwide population-based study of C611Y carriers.

Consequently, we conducted the largest genotype–phenotype study on C611Y carriers to date, focusing exclusively on the C611Y variant, aiming to establish a risk profile for optimal clinical management.

## 2. Methods

### 2.1. Study Design and Setting

This investigation was a nationwide study of all cases (n = 128) born after 1 January 1930 and recognized as carrying the *RET* C611Y variant in Denmark before 1 April 2021 [2].

### 2.2. Data Sources

The Danish MEN 2 cohort 1901–2021 was used [21]. Once cases had been recognized as C611Y carriers, data were recorded both retrospectively and prospectively.

### 2.3. Participants

Between 1 January 1901 and 1 April 2021, 263 cases (250 MEN 2A, 13 MEN 2B) were registered in the Danish MEN 2 cohort [21]. Of these, 34 were born in 1901–1929 and excluded [21]. Among the remaining 229 cases, 13 had MEN 2B and were also excluded. This provided a Danish MEN 2A cohort for 1930–2021 with 216 cases. Of these, 88 carried other variants than the *RET* 611Y. Consequently, the Danish *RET* C611Y cohort for 1930–2021 contained 128 cases (Figure 1).

### 2.4. Variables

An index case was defined as a clinically affected individual through whom attention was first drawn to the presence of MEN 2A in a family [24]. In index cases, the date of MEN 2A diagnosis was recorded as the date where suspicion of MEN 2A was raised for the first time in the medical record. In non-index cases, it was recorded as the date of a positive *RET* test or the date of prophylactic thyroid surgery, according to whichever came first. MEN 2A cases (n = 6) in which the carrier was unaware of the syndrome before death or emigration were omitted from calculations of age at MEN 2A diagnosis.

Normal thyroid, C-cell hyperplasia (CCH), MTC, PHEO, CLA, and Hirschsprung’s disease were defined by histology. The date of surgery was recorded as the date of diagnosis. If diagnosis was conducted at autopsy, date of death was recorded as the date of diagnosis.

Prophylactic thyroidectomy was defined, in accordance with the 2015 American Thyroid Association (ATA) guidelines on MTC, as the removal of the thyroid before MTC develops or while it is clinically unapparent and confined to the gland [25], or as thyroidectomy in a clinically asymptomatic or presymptomatic carrier [26,27]. MTC staging was performed according to the AJCC TNM staging system (seventh and eighth edition) [28,29]. Distant metastases were considered by imaging and/or histology. A biochemical cure was established by basal serum calcitonin levels below the limit of detection or the limit of quantification at last biochemical follow-up [30].

For the assessment of the age-related progression of MTC, we created four groups: Group 1, normal thyroid/CCH, included carriers with a normal thyroid or CCH and the absence of MTC at initial thyroid surgery. Group 2, MTC T1-4N0M0, included carriers with MTC and no regional or distant metastases at initial thyroid surgery. Group 3, MTC TxN1M0, included carriers with regional lymph node metastases at initial thyroid surgery or any time during follow-up. Group 4, MTC TxNxM1, included carriers with distant metastases at initial thyroid surgery or any time during follow-up.

PHPT was defined as hypercalcemia and an elevated or inappropriately normal parathyroid hormone level [31]. Cured PHPT was defined as the reestablishment of normal calcium homeostasis lasting for a minimum of six months after parathyroidectomy. PHPT was considered to have persisted upon the failure to achieve normocalcemia within six months of parathyroidectomy, while recurrence was defined as the recurrence of hypercalcemia after a normocalcemic interval of more than six months after parathyroidectomy [32].

### 2.5. Statistical Analysis

Continuous data were displayed as median and range or mean and confidence interval where appropriate. Categorical data were shown with absolute and relative values. Cuzick’s test was used to evaluate trends in the age-related progression of MTC. The Kaplan–Meier method was used for estimating age-related penetrance and overall survival. All tests were two-sided, and *p*-values < 0.05 were considered statistically significant. All analyses were conducted using Stata 17.0 (StataCorp, College Station, TX, USA).

### 2.6. Ethics

The investigation was approved by the Region of Southern Denmark (21/20832) and the Danish Data Protection Agency (21/31449).

## 3. Results

### 3.1. Demographics

Overall characteristics of the 128 included *RET* C611Y carriers are shown in Table 1. The female–male ratio was 0.75. Median follow-up after birth was 47 years (range, 3–92). De novo variants were not observed.

### 3.2. Thyroid

The characteristics of carriers who had undergone thyroidectomy are shown in Table 2. The distribution across the TNM stages for the overall group was 0 (41%), I (40%), II (2%), III (12%), IVa (5%), IVb (0%), and IVc (2%). Comparison of distribution among index cases and non-index cases showed significantly lower stages in the non-index group (<0.001) (Fischer’s exact test).

Analysis of the age-related progression of MTC is seen in Table 3. The age-related penetrance of MTC at 10 years was 0%, at 20 years was 4% (CI, 1–12), at 30 years was 18% (CI, 11–29), at 40 years was 50% (CI, 38–62), at 50 years was 79% (CI, 68–88), at 60 years was 86% (CI, 76–93), and at 70 years was 98% (CI, 91–100) (Figure 2).

### 3.3. PHEO

The characteristics of carriers with PHEO are presented in Table 4.

The age-related penetrance of PHEO at 20 years was 0%, at 30 years was 2% (CI, 1–8), at 40 years was 9% (CI, 5–18), at 50 years was 17% (CI, 10–27), at 60 years was 22% (CI, 14–34), and at 70 years was 24% (CI, 16–37) (Figure 3).

### 3.4. PHPT

Among the 128 carriers, data on calcium level was available in 122 cases. The characteristics of those with PHPT are depicted in Table 5.

The age-related penetrance of PHPT at 20 years was 0%, at 30 years was 1% (CI, 0.2–7), at 40 years was 4% (CI, 1–11), at 50 years was 8% (CI, 4–17), at 60 years was 10% (CI, 5–20), and at 70 years was 10% (CI, 5–20) (Figure 4).

### 3.5. Other Manifestations

Neither CLA nor Hirschsprung’s disease were observed in this cohort.

### 3.6. Survival

Overall survival at 30 years was 100%, at 40 years was 99% (CI, 91–100), at 50 years was 93% (CI, 84–97), at 60 years was 85% (CI, 73–91), at 70 years was 74% (CI, 59–84), and at 80 years was 65% (CI, 47–79) (Figure 5).

## 4. Discussion

In this national study of the largest *RET* C611Y cohort to date, we provided a detailed description of disease manifestations (MTC, PHEO, PHPT, CLA, and Hirschsprung’s disease) and survival.

### 4.1. Limitations

For full thyroid data, all carriers need to undergo a total thyroidectomy. In our cohort, 25 carriers had not undergone thyroid surgery at last follow-up. The reasons for this were normal calcitonin (n = 13), MEN 2 diagnosis unknown by the carrier (n = 5), refusal of MEN 2 work-up (n = 4), age younger than recommended for calcitonin screening (n = 2) [25], and carrier preference (n = 1). Thyroidectomy in these cases would be unethical and opposed to the guidelines [25].

PHPT data at last follow-up were missing in six carriers due to being aged younger than recommended for PHPT screening (n = 3) [25], MEN 2 diagnosis being unknown by the carrier (n = 2), or a missing medical record (n = 1). Given that data were missing in only 2% (n = 3) of relevant carriers, this is likely to be insignificant.

Our definition of PHEO may miss those cases where the carrier was not operated on. This proportion, however, is likely very small, as surgical resection is the cornerstone of therapy [33]. Histological definition also enables the inclusion of PHEO diagnosed by autopsy in carriers without prior surgery or biochemical or imaging work-up.

### 4.2. Demographics

The female–male ratio was comparable with previous studies [10,19]. While age at MEN 2A diagnosis has not previously been reported, median age at last follow-up was similar to a previous study [19]. To date, our study is the only to have reported the de novo variant frequency in C611Y carriers. of the rate of de novo variants found here (0%) is lower than that (9%) reported for MEN 2A overall [34].

### 4.3. Thyroid

MTC prevalence in our study (60%) was comparable to that of previous single- and multicenter studies (53–60%) [10,20]. A Chinese study reported a higher MTC prevalence (100%), which may reflect the higher median age at surgery (40 years) compared to ours (33 years) [19]. The proportion of carriers with normal thyroid or C-cell hyperplasia found here (40%) corresponds well with that of previous studies (40–47%) [10,20]. MTC N0M0 was more frequent in our cohort (41% vs. 21–27%), while MTC N1M0 was more frequent in other cohorts (17% vs. 27–32%) [10,20]. The low rate of distant metastases at time of initial surgery found here (2%) is similar to that seen in other studies (0–7%) [10,19,20]. The median follow-up after thyroidectomy in our study is significantly longer (20 years) than reported in a comparable study (0.33 years) [19]. Meanwhile, the cure rate found here (56%) corresponds well with that of the other study (55%) [19]. A higher cure rate (74%) was seen in a German-led multicenter study. An explanation for this discrepancy may be that the study included not only C611Y carriers but also carriers of the C611W and C611F variants, which may be phenotypically less aggressive. Also, the study did not explicitly define a cure or describe the duration until follow-up [10]. The age-related progression of MTC in C611Y carriers has only been reported once previously. Although our results are based on nationwide long-term follow-up and previous results were based on time at thyroidectomy in two single centers, the results are very similar [20]. The age-related penetrance of MTC has never been reported for C611Y carriers alone. However, when comparing with the multicenter study that included all codon 611 variant carriers, the age-related penetrance of MTC found here is only slightly higher than that seen in the multicenter study (34). This may reflect a difference in populations.

### 4.4. PHEO

The prevalence of PHEO (15%) found in our study corresponds well with that (12%) reported in a recent Chinese study [19] but is somewhat lower than that (27%) reported in a multicenter study by the International *RET* Exon 10 Consortium [10]. As our study was much larger (n = 128 vs. n = 15) and without selection bias, it is likely that the prevalence of PHEO is lower than that reported by the Exon 10 Consortium. The median age at PHEO diagnosis (42 years) in our cohort is markedly lower than that (53 years) seen in the Chinese study. However, the data from that study were only based on two C611Y carriers with PHEO [19]. We report for the first time on C611Y carriers with hypertensive crisis (11%) and malignant PHEO (5%), documenting that, although rare, this may occur. Also, we report for the first time on the C611Y-specific age-related penetrance of PHEO. The age-related penetrance found here largely corresponds with that found in two large studies looking at carriers of all 611 variants [10,35].

### 4.5. PHPT

The PHPT prevalence found in our study (7%) was comparable to that of previous studies (0–7%) [10,19]. We report for the first time C611Y-specific data on age at diagnosis, biochemistry, symptoms, hypercalcemic crisis, surgical procedure, histology, and follow-up for PHPT, hindering literature comparison. Similarly, we report on the C611Y-specific age-related penetrance of PHPT for the first time. However, the age-related penetrance found here is comparable to that reported on all 611 variant carriers by the International *RET* Exon 10 Consortium [10].

### 4.6. Other Manifestations

In our cohort, there were no carriers with CLA. This may to some extent be explained by our strict histological definition but likely represents real-world reality, as this manifestation has only been reported once in the C611Y literature [19]. Also, we found no carriers with Hirschsprung’s disease, pointing towards an extremely low likelihood of this manifestation in C611Y carriers. Similar results have been found in several smaller studies [10,19,36].

### 4.7. Survival

Our survival data are the first ever reported on C611Y carriers and among very little long-term survival data on MEN 2A in general. This hinders one-on-one comparison to the literature. However, if conducting a rough comparison with a large Danish reference population from another nationwide population-based study with a roughly similar female–male ratio, the overall survival of C611Y carriers seems comparable to that of the general population [37].

## 5. Conclusions

*RET* C611Y is associated with a very high penetrance of MTC and a low penetrance of PHEO and PHPT. CLA and Hirschsprung’s disease almost never occur. MTC seems moderately aggressive, but large variability can be seen. Overall survival may be comparable to that of the general population.

## Figures and Tables

**Figure 1 cancers-17-00374-f001:**
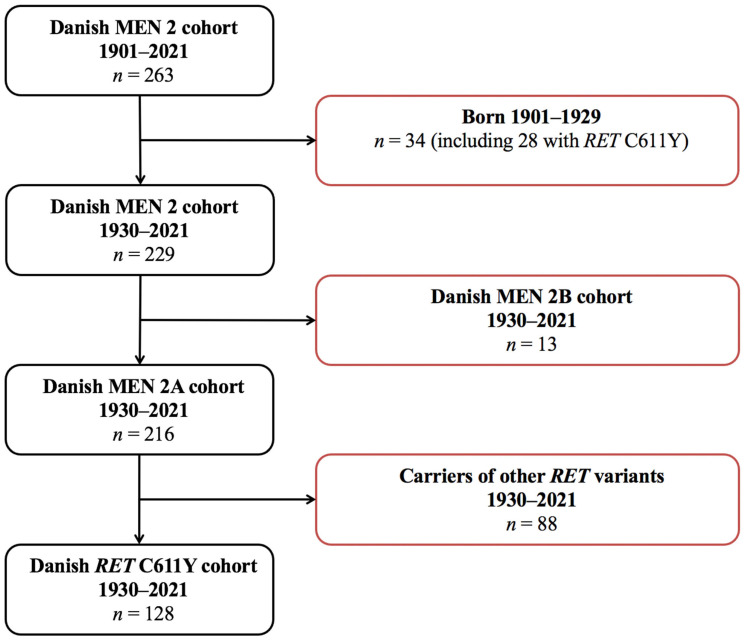
Danish *RET* C611Y cohort for 1930–2021. Abbreviations: MEN, multiple endocrine neoplasia; *RET*, rearranged during transfection.

**Figure 2 cancers-17-00374-f002:**
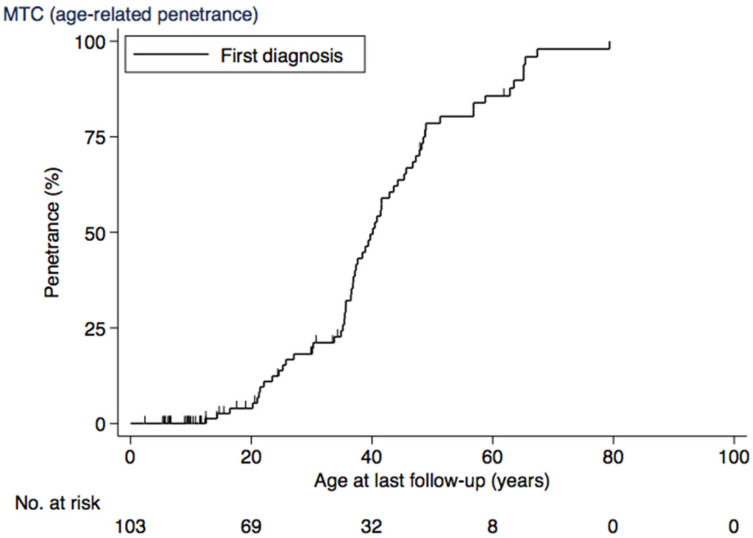
Age-related penetrance of MTC. Abbreviations: MTC, medullary thyroic cancer.

**Figure 3 cancers-17-00374-f003:**
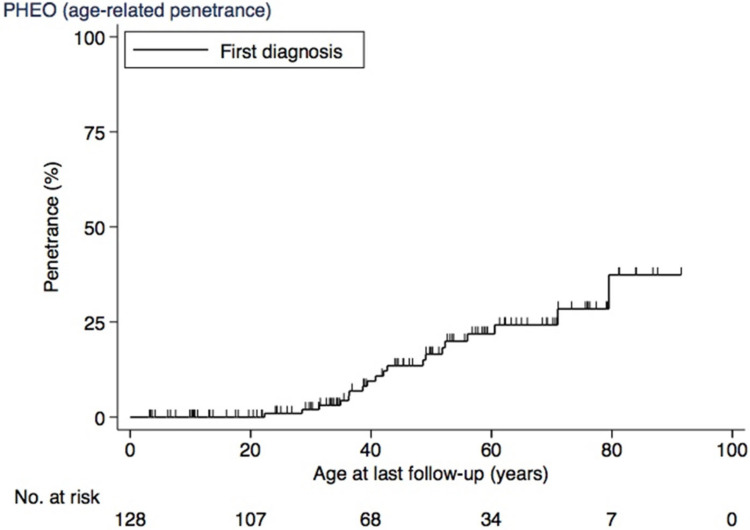
Age-related penetrance of PHEO. Abbreviations: PHEO, pheochromocytoma.

**Figure 4 cancers-17-00374-f004:**
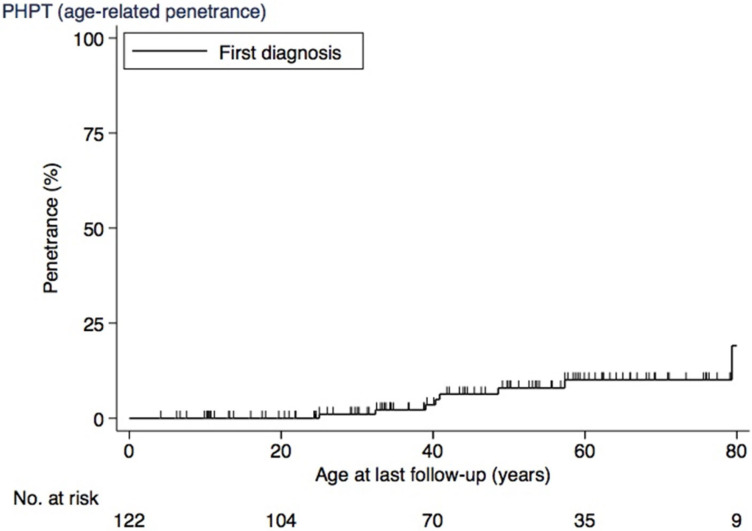
Age-related penetrance of PHPT. Abbreviations: PHPT, primary hyperparathyroidism.

**Figure 5 cancers-17-00374-f005:**
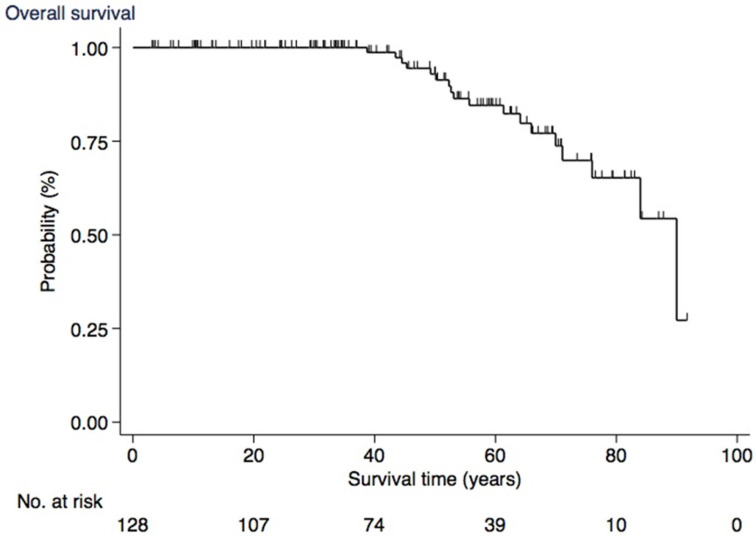
Overall survival.

**Table 1 cancers-17-00374-t001:** *RET* C611Y carriers in Denmark 1930–2021.

	All	Index	Non-Index	
Characteristics	(n = 128)	(n = 10)	(n = 118)	*p* Value
Sex, n (%)				0.325 ^a^
Female	55 (43)	6 (60)	49 (42)	
Male	73 (57)	4 (40)	69 (58)	
Age at MEN 2A diagnosis,				0.053 ^b^
median years (range)	25 (0.1–79) ^c^	37 (22–68) ^d^	24 (0.1–79) ^e^	
Thyroid, n (%)				0.151 ^a^
No surgery	25 (20)	1 (10)	24 (20)	
Normal/CCH	41 (32)	1 (10)	40 (34)	
MTC	62 (48)	8 (80)	54 (46)	
PHEO, n (%)				0.041 ^a^
Yes	19 (15)	4 (40)	15 (13)	
No	109 (85)	6 (60)	103 (87)	
PHPT, n (%) ^c^				0.130 ^a^
Yes	8 (7)	2 (20)	6 (5)	
No	114 (93)	8 (80)	106 (95)	

Abbreviations: *RET*, rearranged during transfection; MEN 2A, multiple endocrine neoplasia type 2A; CCH, C-cell hyperplasia; MTC, medullary thyroid cancer; PHEO, pheochromocytoma; PHPT, primary hyperparathyroidism; CLA, cutaneous lichen amyloidosis.^a^ Fischer’s exact test. ^b^ Wilcoxon rank sum test. ^c^ Based on 122 cases with pertinent data. ^d^ Based on 9 cases with pertinent data. ^e^ Based on 113 cases with pertinent data.

**Table 2 cancers-17-00374-t002:** Thyroid in *RET* C611Y carriers in Denmark 1930–2021.

	All	Index	Non-Index	
Characteristics	(n = 103)	(n = 9)	(n = 94)	*p* Value
At first thyroidectomy Sex, n (%)				0.478 ^a^
Female	41 (40)	5 (55)	36 (38)	
Male	62 (60)	4 (44)	58 (62)	
Age at thyroidectomy,				0.039 ^b^
median years (range)	33 (2–79)	38 (30–67)	30 (2–79)	
Thyroidectomy, n (%)				<0.001 ^a^
Tumorectomy	1 (1)	1 (11)	0	
HT	1 (1)	1 (11)	0	
ST	1 (1)	1 (11)	0	
TT	100 (97)	6 (67)	94 (100)	
Prophylactic, n (%)				<0.001 ^a^
Yes	96 (93)	3 (33) ^c^	93 (99)	
No	7 (7)	6 (67)	1 (1)	
Lymph node surgery, n (%)				<0.001 ^a^
None	53 (51)	2 (22)	51 (54)	
Selective	12 (12)	2 (22)	10 (11)	
CND	26 (25)	3 (33)	23 (24)	
LND	2 (2)	1 (11)	1 (1)	
CND + LND	10 (10)	1 (11)	9 (10)	
Pathology				<0.001 ^a^
Normal/C-cell hyperplasia	41 (40)	1 (11)	40 (43)	
MTC N0M0	42 (41)	2 (22)	40 (43)	
MTC N1M0	18 (17)	4 (44)	14 (15)	
MTC NxM1	2 (2)	2 (22)	0	
T category, n (%)				<0.001 ^a^
Tx	1 (1)	1 (11)	0	
T0	41 (40)	1 (11)	40 (43)	
T1a	34 (33)	1 (11)	33 (35)	
T1b	19 (18)	5 (56)	14 (15)	
T2	7 (7)	0	7 (7)	
T3	0	0	0	
T4a	1 (1)	1 (11)	0	
T4b	0	0	0	
N category, n (%)				<0.001 ^a^
N0	83 (81)	3 (33)	80 (85)	
N1a	12 (12)	2 (22)	10 (11)	
N1b	8 (8)	4 (44)	4 (4)	
M category, n (%)				0.007 ^a^
M0	101 (98)	7 (78)	94 (100)	
M1	2 (2)	2 (22)	0	
At last follow-up				
Follow-up after thyroidectomy	20 (0.5–49)	12 (0.5–49)	20 (1–43)	0.248 ^b^
median years (range)				
Status, n (%)				0.010 ^a^
Cure	58 (56)	1 (11)	57 (61)	
No cure	45 (44)	8 (89)	37 (39)	

Abbreviations: *RET*, rearranged during transfection; HT, hemithyroidectomy; ST, subtotal thyroidectomy; TT, total thyroidectomy; CND, central neck dissection; LND, lateral neck dissection. ^a^ Fischer’s exact test. ^b^ Wilcoxon rank sum test. ^c^ All three patients initially presented with pheochromocytoma.

**Table 3 cancers-17-00374-t003:** Age-related progression of MTC in *RET* C611Y carriers in Denmark 1930–2021.

		MTC			
	Normal/CCH	T1-4N0M0	TxN1M0	TxNxM1	
	(n = 41)	(n = 42 ^b^)	(n = 20 ^c^)	(n = 7)	*p* Value
At first diagnosis					
Age,					<0.001 ^a^
mean years (CI)	16 (12–20)	38 (34–42)	45 (38–53)	49 (36–61)	

Abbreviations: MTC, medullary thyroid carcinoma; *RET*, rearranged during transfection; CCH, C-cell hyperplasia; CI, confidence interval. ^a^ Cuzick’s test for trend. ^b^ From the initial T1-4N0M0 group, 1 case developed N1 and later M1, 1 case developed N1 only, and 1 case developed M1 during follow-up. ^c^ From the initial TxN1M0 group, 3 cases develop M1 during follow-up.

**Table 4 cancers-17-00374-t004:** PHEO in *RET* C611Y carriers in Denmark 1930–2021.

	All	Index	Non-Index	
Characteristics	(n = 19)	(n = 4)	(n = 15)	*p* Value
At PHEO diagnosis				
Sex, n (%)				1 ^a^
Female	9 (47)	2 (50)	7 (47)	
Male	10 (53)	2 (50)	8 (53)	
Age, median years (range)	42 (22–79)	38 (22–49)	43 (29–79)	0.194 ^b^
Symptoms, n (%)				0.255 ^a^
Yes	13 (68)	4 (100)	9 (60)	
No	6 (32)	0	6 (40)	
Hypertensive crisis, n (%)				0.386 ^a^
Yes	2 (11)	1 (25)	1 (7)	
No	17 (89)	3 (75)	14 (93)	
ADX, n (%)				0.648 ^a^
None ^c^	2 (11)	1 (25)	1 (7)	
Cortical sparing	2 (11)	0	2 (13)	
Total	15 (79)	3 (75)	12 (80)	
Largest tumor size,				0.029 ^b^
median mm (range)	26 (13–150) ^d^	50 (30–150)	22 (13–65) ^e^	
Laterality, n (%)				1^a^
Unilateral	18 (95)	4 (100)	14 (93)	
Bilateral	1 (5)	0	1 (7)	
Metastatic, n (%)				0.211 ^a^
Yes	1 (5)	1 (25)	0	
No	18 (95)	3 (75)	15 (100)	
At last follow-up ^f^				
Follow-up after first ADX,				0.457 ^b^
median years (range)	12 (0.5–55)	17 (0.6–30)	11 (0.5–55)	
Status, n (%)				1 ^a^
No further surgery	16 (89)	4 (100)	12 (86)	
Contralateral PHEO	2 (11) ^g^	0	2 (14) ^g^	
Ipsilateral recurrence	0	0	0	

Abbreviations: PHEO, pheochromocytoma; *RET*, rearranged during transfection; ADX, adrenalectomy. ^a^ Fischer’s exact test. ^b^ Wilcoxon rank sum test. ^c^ One non-index case was diagnosed at autopsy and one index case was diagnosed by coarse needle biopsy. ^d^ Based on 18 cases with pertinent data. ^e^ Based on 14 cases with pertinent data. ^f^ Based on 18 cases, as one case was diagnosed at autopsy. ^g^ Time to ADX for contralateral PHEO was 2 and 13 years.

**Table 5 cancers-17-00374-t005:** PHPT in *RET* C611Y carriers in Denmark 1930–2021.

	All	Index	Non-Index	
Characteristics	(n = 8)	(n = 2)	(n = 6)	*p* Value
At PHPT diagnosis				
Sex, n (%)				0.464 ^a^
Female	2 (75)	1 (50)	1 (17)	
Male	6 (25)	1 (50)	5 (83)	
Age, median years (range)	41 (24–79)	44 (39–49)	41 (24–79)	1 ^b^
Ionized calcium level,				0.051 ^b^
median mmol/L (range)	1.37 (1.34–1.65) ^c^	1.59 (1.53–1.65)	1.37 (1.34–1.38) ^d^	
PTH level,				0.046 ^b^
median pmol/L (range)	12 (3–19)	18 (16–19)	9 (3–15)	
Symptoms, n (%)				1 ^a^
Yes	3 (38)	1 (50)	2 (33)	
No	5 (63)	1 (50)	4 (67)	
Hypercalcemic crisis, n (%)				
Yes	0	0	0	
No	8 (100)	2 (100)	6 (100)	
PTX, n (%)				0.429 ^a^
None	2 (25)	0	2 (33)	
Selective	2 (25)	0	2 (33)	
Subtotal	4 (50)	2 (100)	2 (33)	
Histology, n (%) ^e^				0.467 ^a^
Adenoma	4 (67)	2 (100)	2 (50)	
Hyperplasia	2 (33)	0	2 (50)	
Carcinoma	0	0	0	
At last follow-up ^e^				
Follow-up after first PTX,				0.165 ^b^
median years (range)	20 (0.3–28)	6 (0.3–12)	28 (3–28)	
Status, n (%)				
Cure	6 (100)	2 (100)	4 (100) ^f^	
Persistence	0	0	0	
Recurrence	0	0	0	

Abbreviations: PHPT, primary hyperparathyroidism; *RET*, rearranged during transfection; PTH, parathyroid hormone; PTX, parathyroidectomy. ^a^ Fischer’s exact test. ^b^ Wilcoxon rank sum test. ^c^ Based on 7 cases. The last case had total calcium level measured. ^d^ Based on 5 cases. The last case had total calcium level measured. ^e^ Based on 6 cases who underwent PTX. ^f^ One case experienced recurrence 20 years after first PTX. Cure was reestablished after second PTX.

## Data Availability

Data collected for this study will be made available by the corresponding author upon reasonable request.

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
