# Peer review of "RET C611Y Germline Variant in Multiple Endocrine Neoplasia Type 2A in Denmark 1930–2021: A Nationwide Study"

_cancers, 2025, doi:10.3390/cancers17030374_

Round 1
Reviewer 1 Report
Comments and Suggestions for Authors
In this manuscript, the authors present a nationwide study of 10 index cases with the RET C611Y change, as well as 118 relatives carrying this change. This change seems to be frequent in Denmark (128 out of 216) Authors provide a detailed description of the clinical manifestations (MTC, PHEO, PHPT, CLA and Hirschsprung’s disease) in all these families and survival.
Previously, families with this C611Y change had already been described, but studies included fewer cases.
Minor change:
Figure 1. Instead of: “Not RET C611Y carriers”, I think it would be better to change to “carriers for other pathogenic variants in RET”
Comments:
1.- The fact that this change is so prevalent in Denmark, the authors suggest that it may be due to a founder effect. However, they have not performed a haplotype study, so it is possible that not all families come from a common ancestor.
2.- it is somewhat striking that none of the patients present CLA or Hirschsprung's disease.
3.- No one of 10 index cases was a “de novo” variant. Prevalence, as authors mention, is about 9%, therefore in 10 index cases is not surprising that no “de novo” cases have been found. I guess family members should not be included.
The article is well written and structured.
Author Response
MINOR CHANGE: Figure 1. Instead of: “Not RET C611Y carriers”, I think it would be better to change to “carriers for other pathogenic variants in RET”
RESPONSE: We agree with your comment and have edited Figure 1 accordingly.
COMMENT 1: The fact that this change is so prevalent in Denmark, the authors suggest that it may be due to a founder effect. However, they have not performed a haplotype study, so it is possible that not all families come from a common ancestor.
RESPONSE 1: Discussing this matter in text (line 75-77 in the revised manuscript), we reference a study conducted by Mathiesen et al (ref. 23) also based on the Danish MEN 2A cohort. The relatedness coefficient between individuals from 12 Danish RET C611Y families was estimated using single nucleotide polymorphism data. According to global relatedness coefficients, none of the families were closely related. However, when estimating the local relatedness at the RET locus, all 12 apparently unrelated C611Y families had a high probability of being related and sharing a common haplotype. Therefore, we find it likely that the high prevalence of the RET C611Y variant observed in Denmark is due to a founder effect.
COMMENT 2: It is somewhat striking that none of the patients present CLA or Hirschsprung's disease.
RESPONSE 2: Yes, these manifestations seem to be rare for the RET C611Y variant as CLA only has been reported once in the literature [19] and Hirschprung’s disease never has been reported.
- Qi XP, Peng JZ, Yang XW, et al. The RET C611Y mutation causes MEN 2A and associated cutaneous. Endocr Connect. Sep 1 2018;7(9):998-1005. doi:10.1530/EC-18-0220
COMMENT 3: No one of 10 index cases was a “de novo” variant. Prevalence, as authors mention, is about 9%, therefore in 10 index cases is not surprising that no “de novo” cases have been found. I guess family members should not be included.
RESPONSE 3: We agree, it was indeed not surprising that no de novo variants were observed.
Reviewer 2 Report
Comments and Suggestions for Authors
I have carefully read the manuscript entitled “RET C611Y germline variant in multiple endocrine neoplasia type 2A in Denmark 1930-2021”, which reports the largest nationwide epidemiological data of RET C611Y carriers to date. This manuscript is well written and provides relevant information on the epidemiology and phenotypic presentation of the MEN2 C611Y carriers. This work has raised also some non-critical questions and remarks, which are indicated below :
1) Introduction, line 59, could the authors indicate the OMIM number for MEN2A #171400
2) Methods, line 90: 34 patients born between 1901 and 1929 were excluded. Can the authors explain this choice? Too many missing data before 1930? Different management from the rest of the cohort?
3) Results, table 1: Insofar as no patient presented with a de novo variant, no CLA or HSCR, it seems possible to remove these data from the table in order to make it more reader-friendly. These data could only be presented in the text.
4) Table 2: I'm rather surprised by the data on prophylactic surgery. According to the definition given in the methodology, prophylactic surgery is defined as removal of the thyroid before MTC develops or while it is clinically unapparent and confined to the gland, or as thyroidectomy in clinically asymptomatic or presymptomatic carrier.
- In index cases, the authors reported 3 prophylactic surgeries. Does that mean that 3 patients had pheochromocytoma or primary hyperparathyroidism as the first MEN2-associated manifestation? Or are these prophylactic surgeries related to localized MTC to the thyroid gland? If available, could the authors provide presurgical calcitonin and CEA values, and preoperative ultrasound data for both index and non-index cases.
- In non-index cases, the authors reported 93 (99%) prophylactic thyroidectomy, but 14 (15%) exhibited pTxN1 stage at time of diagnosis. Moreover, 32 underwent central neck dissection and 10 lateral neck dissection, with evidence of lymph node invasion in 10/32 (33%) and 4/10 (40%), respectively. Did calcitonin or preoperative ultrasound suggest locally advanced disease? In this case, patients probably have locally advanced disease at diagnosis and it seems difficult to consider these surgeries as prophylactic.
5) Table 2: among patients who have not been cured (n=45), can the authors specify which patients have biochemical persistence and which have structural disease?
6) Table 4: Of the 15 non-index cases who developed pheochromocytoma, only one exhibited a bilateral disease, but it is seems 2 have a contralateral pheochromocytoma. Could the authors clarify this point.
7) Results, line 215, survival: Do the authors have any information on specific survival, in particular for metastatic MTC and pheochromocytoma?
8) Discussion, line 227: I'm not sure I understand the following sentence: “In our cohort, 25 carriers had not undergone thyroid surgery […], MEN2 diagnosis unknow for the carrier (n=5)”. In this case, how can they be considered as carrier? Does this mean that there are 5 patients who have not been genetically screened? Please the authors clarify this point.
9) Overall, tables take up a lot of space and the body of the text is buried in the middle, making it difficult to read, particularly concerning the results.
Author Response
COMMENT 1: Introduction, line 59, could the authors indicate the OMIM number for MEN2A #171400
RESPONSE 1: Yes.
COMMENT 2: Methods, line 90: 34 patients born between 1901 and 1929 were excluded. Can the authors explain this choice? Too many missing data before 1930? Different management from the rest of the cohort?
RESPONSE 2: Inclusion was chosen from 1 January 1930, under the assumption that virtually all patients with MEN 2A in Denmark born after this date have been captured in the cohort. This is explained in the citation presented below from Holm et al (Page 3, section “Materials and Methods”, point “2.3 Participants”), which is another study based on the Danish MEN 2A cohort. To make this clear for our study, we have added the reference to the revised manuscript at the end of the following sentence in the Methods section line 90: “Of these, 34 were born in 1901-1929 and excluded [21]”.
“Inclusion was chosen from 1 January 1930, under the assumption that virtually all patients with MEN 2A in Denmark born after this date have been captured in the cohort. Since MTC usually is the first manifestation of MEN 2A and node-negative MTC often develops between the third to fifth decade of life, practically all patients with MTC as part of MEN 2A should have been recognized in the Danish MTC cohort 1960–2014. Additionally, the incidence of MEN 2A in Denmark has been roughly stable since the 1930s, arguing that virtually all MEN 2A patients since then have been captured.”[21]
- Holm M, Vestergaard P, Poulsen MM, et al. Primary Hyperparathyroidism in Multiple Endocrine Neoplasia Type 2A in Denmark: A Nationwide Population-Based Retrospective Study in Denmark 1930-2021. Cancers (Basel). Apr 2 2023;15(7)doi:10.3390/cancers15072125
COMMENT 3: Results, table 1: Insofar as no patient presented with a de novo variant, no CLA or HSCR, it seems possible to remove these data from the table in order to make it more reader-friendly. These data could only be presented in the text.
RESPONSE 3: We agree with your comment and have removed these variables from table 1 and rather comment on these variables in the text.
COMMENT 4, PART 1: Table 2: I'm rather surprised by the data on prophylactic surgery. According to the definition given in the methodology, prophylactic surgery is defined as removal of the thyroid before MTC develops or while it is clinically unapparent and confined to the gland, or as thyroidectomy in clinically asymptomatic or presymptomatic carrier.
- In index cases, the authors reported 3 prophylactic surgeries. Does that mean that 3 patients had pheochromocytoma or primary hyperparathyroidism as the first MEN2-associated manifestation? Or are these prophylactic surgeries related to localized MTC to the thyroid gland?
RESPONSE 4, PART 1: All three index cases who underwent prophylactic thyroidectomy initially presented with pheochromocytoma. As this information might be interesting to the reader, we have decided to add it to a footnote in table 2.
COMMENT 4, PART 2: If available, could the authors provide presurgical calcitonin and CEA values, and preoperative ultrasound data for both index and non-index cases.
RESPONSE 4, PART 2: Unfortunately, data regarding preoperative ultrasound is incomplete. We agree, information regarding presurgical calcitonin and CEA values is very interesting. However, this is beyond the scope of our study. These data will be presented separately in an exciting ongoing study which soon will be ready for publication.
COMMENT 4, PART 3: In non-index cases, the authors reported 93 (99%) prophylactic thyroidectomy, but 14 (15%) exhibited pTxN1 stage at time of diagnosis. Moreover, 32 underwent central neck dissection and 10 lateral neck dissection, with evidence of lymph node invasion in 10/32 (33%) and 4/10 (40%), respectively. Did calcitonin or preoperative ultrasound suggest locally advanced disease? In this case, patients probably have locally advanced disease at diagnosis and it seems difficult to consider these surgeries as prophylactic.
RESPONSE 4, PART 3: Lateral neck dissection was performed in patients asymptomatic of MTC when preoperative ultrasound suggested more advanced disease. According to the 2015 revised American Thyroid Association (ATA) guidelines for the management of MTC by Wells et al (Recommendation 24 page 582), total thyroidectomy and central neck dissection is recommended in patients with MTC and no evidence of neck lymph node metastases by ultrasound examination and no evidence of distant metastases [25].
We acknowledge that the true prophylactic nature of the surgeries can be questioned. The explanation lies in our rather broad definition of prophylactic thyroidectomy. This definition is recognized and widely used in the field of MEN 2 [25,26,27]. We consider these surgeries prophylactic as they fall into our definition of removal of the thyroid before MTC develops or while it is clinically unapparent and confined to the gland [25] or as thyroidectomy in a clinically asymptomatic or presymptomatic carrier [26,27].
- Wells SA, Jr., Asa SL, Dralle H, et al. Revised American Thyroid Association guidelines for the management of medullary thyroid carcinoma. Thyroid. Jun 2015;25(6):567-610. doi:10.1089/thy.2014.0335
- Castinetti F, Moley J, Mulligan L, Waguespack SG. A comprehensive review on MEN2B. Endocrine-related cancer. Feb 2018;25(2):T29-t39. doi:10.1530/erc-17-0209
- Raue F, Frank-Raue K. Update on Multiple Endocrine Neoplasia Type 2: Focus on Medullary Thyroid Carcinoma. Journal of the Endocrine Society. Aug 1 2018;2(8):933-943. doi:10.1210/js.2018-00178
COMMENT 5: Table 2: Among patients who have not been cured (n=45), can the authors specify which patients have biochemical persistence and which have structural disease?
REPONSE 5: According to our definition of biochemical cure, all of the 45 patients that you refer to in table 2 have biochemical persistence/relapse We agree that information regarding structural disease is very interesting. Unfortunately, we do not have such data available. Only 10 patients had postoperative serum calcitonin levels above 150pg/ml and thereby met the ATA recommendations (recommendation 47 and 48, page 590) for more extensive imaging besides ultrasound [25].
- Wells SA, Jr., Asa SL, Dralle H, et al. Revised American Thyroid Association guidelines for the management of medullary thyroid carcinoma. Thyroid. Jun 2015;25(6):567-610. doi:10.1089/thy.2014.0335
COMMENT 6: Table 4: Of the 15 non-index cases who developed pheochromocytoma, only one exhibited a bilateral disease, but it is seems 2 have a contralateral pheochromocytoma. Could the authors clarify this point.
RESPONSE 6: Table 4 is subdivided into “At PHEO diagnosis” and “At last follow-up”. At the time of PHEO diagnosis, one patient had bilateral pheochromocytoma. During follow-up, two patients developed an additional pheochromocytoma, both at the contralateral side of the first pheochromocytoma. By underlining “At PHEO diagnosis” and “At last follow-up” in table 4 (and similarly in table 2 and table 5), we hope to make this subdivision clearer to the reader.
COMMENT 7: Results, line 215, survival: Do the authors have any information on specific survival, in particular for metastatic MTC and pheochromocytoma?
RESPONSE 7: We agree that this information is very relevant. Unfortunately, data regarding specific survival is incomplete and we have therefore decided not to rapport on this.
COMMENT 8: Discussion, line 227: I'm not sure I understand the following sentence: “In our cohort, 25 carriers had not undergone thyroid surgery […], MEN2 diagnosis unknow for the carrier (n=5)”. In this case, how can they be considered as carrier? Does this mean that there are 5 patients who have not been genetically screened? Please the authors clarify this point.
RESPONSE 8: All patients are recognized with the RET C611Y variant. These include patients having tested positive for the C611Y variant by genetic screening (n=117, 91%), obligate carriers (n=7, 5%) and patients with medullary thyroid cancer (n=2, 2%) or pheochromocytoma (n=2, 2%) AND relatedness to individuals who are confirmed carriers of the RET C611Y variant. This is explained more thoroughly in the citation below from Mathiesen et al on page 1481 section “MEN 2A criteria” [2]. In order to make this clearer in our study, we have added this reference to the following sentence in line 84-85 of the Methods section, subsection Study design and setting: “This investigation is a nationwide study of all cases (n=128) born after January 1, 1930, and recognized with the RET C611Y variant in Denmark before April 1, 2021 [2].”
“If genetic tested, a MEN2 patient was defined as 1) an individual with a pathogenic RET germline sequence change and a MEN2A phenotype in the ARUP MEN2 database on May 1, 201815. If not genetic tested, a MEN2A patient was defined as 2) an individual with a MEN2A feature (histologically verified MTC/pheochromocytoma/Hirschsprung’s disease or clinically diagnosed cutaneous lichen) and relatedness to an individual fulfilling 1), or 3) an individual without a proven MEN2A feature, but providing linkage between two individuals fulfilling 1), or 4) an individual without a proven MEN2A feature, but providing linkage between a patient with a MEN2A feature and an individual fulfilling 1), or 5) an individual with >1 MEN2A feature.” [2]
- Mathiesen JS, Kroustrup JP, Vestergaard P, et al. Incidence and prevalence of multiple endocrine neoplasia 2A in Denmark 1901-2014: a nationwide study. Clinical epidemiology. 2018;10:1479-1487. doi:10.2147/clep.S174606
COMMENT 9: Overall, tables take up a lot of space and the body of the text is buried in the middle, making it difficult to read, particularly concerning the results.
RESPONSE 9: We agree that tables take up quite a bit of space which potentially makes the manuscript difficult to read. This is a general challenge in studies were many variables are presented. We aim to be as transparent as possible with our dataset, and hope that the presentation of data in tables contributes to this. We also believe that clinicians can benefit from the easy and accessible, yet detailed overview that these tables offer for each of the MEN 2A manifestation. During this revision, we have moved three variables from table 1 to the main text, so this table now take up less space. The current Word format is not ideal. However, as the skilled graphic designers of the editorial office edit the layout of the manuscript into its final form, we believe it will become much more reader-friendly.
Reviewer 3 Report
Comments and Suggestions for Authors
The paper by Hansen et al., titled “RET C611Y Germline Variant in Multiple Endocrine Neoplasia Type 2A in Denmark, 1930–2021,” presents a study conducted on a Danish patient cohort, focusing on the C611Y variant in MEN 2A.
The subject of this paper aligns well with the scope of Cancers. Although the study addresses a relatively niche issue, the questions it explores are relevant (particularly to the Danish population, where the C611Y variant is the most frequent).
All procedures have been clearly described (with good graphical presentation).
The manuscript is presented in an intelligible fashion.
The statistical methods used are well chosen.
The limitations section correctly highlights potential constraints of the study, ensuring transparency and guiding the interpretation of the findings.
References are up to date and appropriate.
In my opinion, this manuscript is suitable for publication in Cancers.
Author Response
Thank you for your comments.
Round 2
Reviewer 2 Report
Comments and Suggestions for Authors
The authors presented a revised version of the manuscript responding to the questions and comments of the reviewers. Please ensure that the changes proposed in response to the reviewers appear in the text.